PERSPECTIVE

# Conceptual Exchanges for Understanding Free-Living and Host-Associated Microbiomes

Catherine A. Pfister,[a] Samuel H. Light,[b] Brendan Bohannan,[c] Thomas Schmidt,[d] Adam Martiny,[e] Nicole A. Hynson,[f] Suzanne Devkota,[g] Lawrence David,[h] Katrine Whiteson[i]

[a]Department of Ecology & Evolution and The Microbiome Center, University of Chicago, Chicago, Illinois, USA
[b]Department of Microbiology & Duchossois Family Institute, University of Chicago, Chicago, Illinois, USA
[c]Environmental Studies and Biology, University of Oregon, Eugene, Oregon, USA
[d]Ecology and Evolutionary Biology, University of Michigan, Ann Arbor, Michigan, USA
[e]Earth System Science & Ecology and Evolutionary Biology, University of California Irvine, Irvine, California, USA
[f]Pacific Biosciences Research Center, University of Hawaii at Manoa, Honolulu, Hawaii, USA
[g]Microbiome Research, F. Widjaja Foundation Inflammatory Bowel and Immunobiology Research Institute, Cedars-Sinai Medical Center, Los Angeles, California, USA
[h]Molecular Genetics & Microbiology, Duke University, Durham, North Carolina, USA
[i]Department of Molecular Biology and Biochemistry, University of California Irvine, Irvine, California, USA

**ABSTRACT** Whether a microbe is free-living or associated with a host from across the tree of life, its existence depends on a limited number of elements and electron donors and acceptors. Yet divergent approaches have been used by investigators from different fields. The "environment first" research tradition emphasizes thermodynamics and biogeochemical principles, including the quantification of redox environments and elemental stoichiometry to identify transformations and thus an underlying microbe. The increasingly common "microbe first" research approach benefits from culturing and/or DNA sequencing methods to first identify a microbe and encoded metabolic functions. Here, the microbe itself serves as an indicator for environmental conditions and transformations. We illustrate the application of both approaches to the study of microbiomes and emphasize how both can reveal the selection of microbial metabolisms across diverse environments, anticipate alterations to microbiomes in host health, and understand the implications of a changing climate for microbial function.

**KEYWORDS** microbiome, host-microbiome, oxidation state, redox, biogeochemistry, microbial metabolisms, stoichiometry

The recognition that species across the tree of life host microbiomes and that microbes are ubiquitous across ecosystems (1) has led to a surge of interest in understanding the role of a microbiome and its constituent members. Established and developing methodologies are revealing contributions of microbes, including vitamin (2) or nutrient provisioning (3, 4) across a variety of hosts, providing evidence for observations almost a hundred years ago suggesting a role for gut bacteria in provisioning vitamins in a rodent gut (5). Squid sensory organ development is dependent upon bacteria (6), and microbes change the scope for adaptation in *Drosophila* (7). In humans, the microbiome is critical to healthy immune development (8), and altered gut microbial composition is associated with inflammatory bowel diseases and irritable bowel syndrome in adults (9–11) and pediatric necrotizing enterocolitis in infants (12). Further, the microbial composition within the gut is associated with certain cancers (13, 14), antitumor activity (15), diabetes, obesity, and allergies (16). Across a broad swath of ecosystems, newly described metabolisms and taxa involving the nitrogen cycle have been detected (17, 18). While microbiomes exist across diverse ecosystems and hosts, there are numerous methodologies that unite these studies (19), as well as common ecological factors that underlie temporal dynamics and environmental forces that determine

Address correspondence to Catherine A. Pfister, cpfister@uchicago.edu.

The authors declare no conflict of interest.

microbial membership (20). Studies of microbes from extreme environments on earth to the inner environment of hosts have common goals, but their study is often approached differently. We summarize these approaches and their synergies and consider future ways for the study of microbes across diverse environments to proceed.

Studies of microbes from an "environment first" perspective rely on the signature of a metabolism in the environment, often based on a biogeochemical measurement. Here, biogeochemistry, chemical disequilibrium, and thermodynamics all contribute to understanding microbial processes. Indeed, the discovery of many microbial metabolisms was initiated based on measures of chemical transformations or environmental variables that provided clues to potential microbial metabolisms (21–24) (see below).

The alternative "microbe first" approach is based on the microbial taxon serving as a biosensor of possible functional properties. Through cultivation or genomic methodologies, taxonomic identity and gene function can be determined. Across systems, there are increasing efforts to directly link genomics and metabolomics, including in studies of humans (25), where the majority of human-associated taxa can be cultured and are identifiable at the level of nucleotide sequence (26, 27). Because many human-microbe interactions are studied in the context of pathogenesis, the isolation of the taxa, the application of Koch's postulates, and the description of the taxa have been the primary focus. It has been comparatively less frequent that investigators have approached the presence of human-hosted bacteria from the perspective of the redox environment, nutrient provisioning, or stoichiometric constraints. Given the development of varied omic methodologies, the "microbe first" approach is increasingly applied across systems to probe microbially mediated ecosystem functions. Regardless of how they are approached, however, there are significant commonalities across all microbial communities. In all host-associated or free-living systems, microbes use electron donors and acceptors and six major elements: carbon, nitrogen, oxygen, sulfur, hydrogen, and phosphorus (28).

The "environment first" approach relies on several methodologies that may have promise in their application to host-associated microbes, including humans. First, the study of environmental microbes has a strong emphasis on element cycling and interdependencies, often referred to as stoichiometry. Second, biogeochemistry strongly frames the study of environmental microbes, where biogeochemistry explicitly incorporates environmental and abiotic measures that are key to describing biology and chemistry. While elemental cycling is a major conceptual theme for free-living microbes, it is less well developed in human microbiome studies and associated therapies. Environmental microbes likely differ in a number of traits as a consequence of not being dependent on a host, either human or otherwise, and biogeochemical constraints are likely important, though little understood. The time scale of the response of environmental microbes may be slower than that of host-associated microbes, whose fitness relies on constant feedback, ephemeral host provisioning, and immunological responses. Hosts respond to microbes in real time, and microbial fitness is dependent on that reciprocity. The different approaches and biological time scales across systems mean that there is the potential to understand adaptations and evolution of microbial function. Direct comparison of host-associated versus free-living taxa, ideally in systems where a microbial taxon can occur across both, is an area where much can be learned, especially about how elements are exchanged between hosts and microbes or cycled within ecosystems. Here, we propose increased integration of "environment first" with "microbe first" approaches across systems to better understand the interactions in host-associated microbiomes, the changes to microbiomes in the human health arena, and implications for free-living microbes in a changing climate (Table 1).

## THE CYCLING OF ELEMENTS AS A LENS FOR METABOLIC DISCOVERY

One of the longest-running inquiries into the biology of lakes in North America was founded over a hundred years ago in Lake Mendota, adjacent to the University of Wisconsin. E. Birge and C. Juday assembled a group of scientists that would make foundational discoveries about all aspects of limnology, from nutrient cycling to the dynamics of trophy fishes. But the earliest studies by Birge and Juday's group were on gas composition in lakes. A 1932

**TABLE 1** Outline of the two approaches, the "environment first" versus the "microbe first"' approach, to understanding host-associated versus environmental microbiomes, with examples of where each approach has been applied separately and in tandem

| Feature | Details for: | |
| --- | --- | --- |
| | Environment first | Microbe first |
| Major discoveries | Anaerobic ammonium oxidation (anammox) | Symbionts, e.g., bobtail squid and their bioluminescent bacteria |
| Principal tools | Stable isotope enrichment, nutrient flux estimates, elemental ratios (stoichiometry), thermodynamics, chemical disequilibrium | Sequencing (metagenomes, metatranscriptomes), microbial culturing |
| Systems where each approach has been applied | River and lake sediments, oxygen minimum zones | Host-associated body sites, large-scale ecosystem surveys |
| A key insight from each approach | External electron transport in the environment | Pathogens involved in disease and in the human gut |
| Exemplars of insights from using both approaches | 1. Coral symbionts, their relation to nitrogen cycling, and their ecosystem effects 2. The human lung and cystic fibrosis | |

study of sediments dredged in Lake Mendota and enclosed in a Pyrex jar (29) yielded a result that puzzled the authors. The apparatus showed a production of $N_2$ gas in excess of what the authors could explain by denitrification. They stated "…it is difficult to explain the origin of the quantity of nitrites or nitrates required to account for the liberated nitrogen. Another alternative is the assumption that elemental nitrogen may be liberated by some other mechanism than that of denitrification. This would probably be regarded as equally unorthodox." Some decades later, Broda (30) considered that based on the principles of evolution and thermodynamics, a bacteria should exist that could utilize $NO_2^-$ as an oxidant to oxidize ammonium gas to dinitrogen gas. He wrote the equation for what would become anaerobic ammonium oxidation or anammox, a process that took almost another 20 years to first quantify in a reactor and then to culture the *Planctomycetes* bacteria responsible (22).

## ELECTRON ACCEPTORS AS DRIVERS OF COMMUNITY COMPOSITION AND METABOLISM IN ANY ENVIRONMENT

The discovery of anammox described above, though remarkable, is an illustration of how microbial discovery via the "environment first" has often proceeded. A process or function was hypothesized based on features of the environment, and only later was a taxon or taxa identified. Throughout microbial discovery in soils, lakes, oceans, and other ecosystems, the favorability of reactions that oxidize or reduce a compound versus reduce it have been a key lens to determining the microbial metabolism that may be operating. Bacteria have a diversity of electron acceptors and donors that they can utilize, and thus, the redox state of the local environment is a controlling feature of the microbial metabolisms and taxa present.

Microbes first appeared on earth before there was considerable oxygen in the atmosphere. In fact, microbes evolved the metabolisms which produced oxygen in the atmosphere. Therefore, microbes possess an enormous diversity of metabolic strategies that do not depend on oxygen as an electron acceptor. While open ocean environments are characterized by oxygen and nitrate as electron acceptors (Fig. 1), ecosystems from soils to soft sediments to the human gut have fewer of these common electron acceptors available. In these environments, external electron transport (EET) is one adaptation to low-oxygen environments. Recognized as a key means of metabolism capable of generating electricity in genera such as *Shewanella* and *Geobacter*, the discovery of external electron transport also originated in aquatic systems, specifically Oneida Lake (31) and the Potomac River (32). Manganic oxide was demonstrated as a terminal electron acceptor in Oneida Lake, and the bacteria in the Potomac oxidized acetate and reduced iron oxide. Additionally, these environmental microbes were shown to utilize electron acceptors outside the cell, termed external electron transport. The recent discovery of external electron transport within the human gut (33) reinforces the idea that understanding redox states and the diversity of electron acceptors and donors across multiple environments provides a template for the taxa and metabolisms expected (Fig. 1). Electrogenic bacteria in

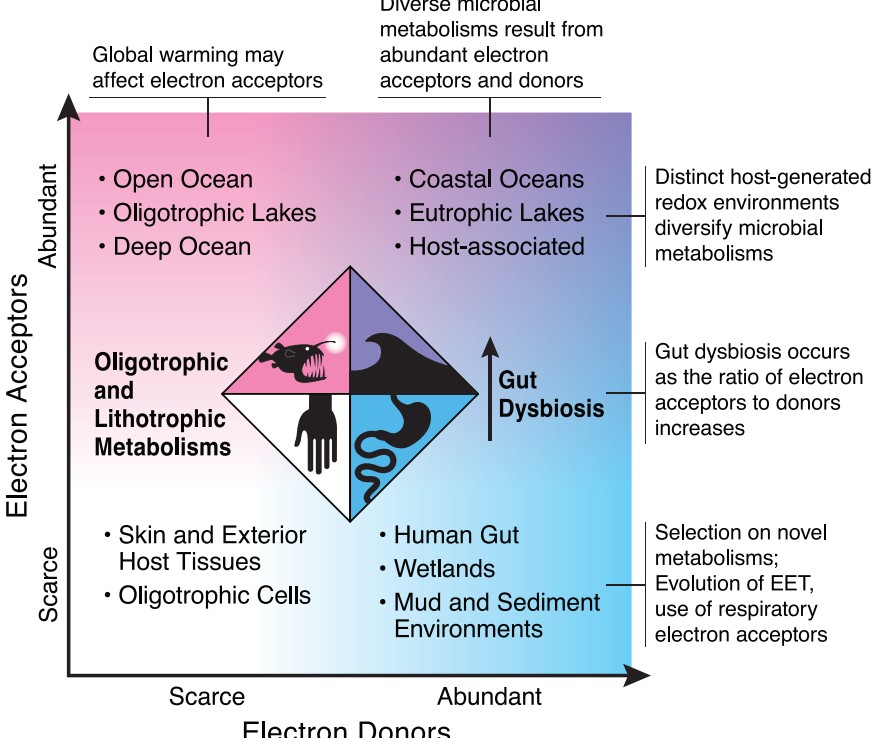

**FIG 1** The portrayal of some habitats classified by whether electron acceptors (red) and donors (blue) for chemotrophic metabolism are relatively scarce or abundant. When electron donors are scarce, lithotrophic metabolisms, or the use of inorganic substrates, occurs. Placing habitats in this simplified diagram illustrates the selective pressures and observations that are common to both host-associated and free-living microbes. It also highlights how changes to a system may be mediated by changes to electron acceptors and donors. For example, gut dysbiosis is associated with an increasing number of electron acceptors. Warming aquatic systems, via climate change, may reduce the distribution of well-oxygenated habitats and thus oxygen as an electron acceptor (figure by Brooks Bays).

humans are likely more common than previously recognized and may be key to mediating the colonization of the pathogenic bacteria *Listeria* in the human gut (33).

Microbial taxa in the human gut are increasingly recognized to use a diverse set of electron acceptors. The ability to access new electron acceptors or induce the host to produce electron acceptors can increase the abundance and the competitive advantage of gut bacteria. *Salmonella* invades the gut, where virulence induces the host to produce a new terminal electron acceptor, tetrathionate (34), allowing the respiring *Salmonella* to compete effectively against taxa relying on fermentation. Similarly, the inflammation in host mice results in nitrate production that serves as an electron acceptor for *Escherichia coli* (35). The prevalent and beneficial human gut microbe, *Faecalibacterium prausnitzii*, uses flavins as a mechanism of transferring electrons externally to oxygen (36), while an opportunistic pathogen in the intestine, *Bilophila wadsworthia*, uses dietary and host-derived organosulfates in bile acids to access sulfite as its electron acceptor. Typically, host bile acids have antimicrobial properties; however, *B. wadsworthia* has subverted this property using taurine-conjugated bile acids to enhance its own fitness (37, 38). As metabolic discovery continues in host-associated microbial taxa, we may conclude that resource use and the availability of electron acceptors are similar and that there are a variety of adaptations across environments to use them. External electron transport demonstrates the capacity for microbes to evolve similar metabolisms across diverse ecosystems and hosts. Across systems, the "environment first" approach can aid in identifying the availability of electron acceptors and how they are altered to predict taxonomic and metabolic change.

## CONTRASTING PATTERNS OF ELECTRON ACCEPTORS AND DONORS ACROSS SYSTEMS

In contrast to mammalian respiratory metabolisms that use few electron donors (sugars, fats, amino acids, etc.) and a single electron acceptor (oxygen), microbial taxa

exploit a range of respiratory electron acceptors and donors to power growth. The diversity of microbial electron donor/acceptor usage is constrained by the environment, whether it is in a water column or an animal gut (Fig. 1). One major distinction in the electron acceptors, donors, and resources available to a microbe is whether it is host associated versus free-living.

While the role of respiration is recognized as critically important in free-living microbes, respiratory metabolisms in the host-microbiome context have received far less attention for several reasons. First, in contrast to the nutritionally poor environments often encountered by free-living microbes, host environments, such as the gut, have abundant energy sources (electron donors) in the forms of carbohydrates, lipids, and more. Fermentative processes thus represent a viable and dominant bioenergetic strategy in this context. Second, inorganic compounds that serve as typical environmental respiratory electron acceptors (nitrate, sulfate, sulfite) tend to be relatively scarce within a healthy gut, though they may play important roles in pathogenesis (37).

Despite the centrality of fermentative processes within the mammalian gut, evidence that unconventional respiratory electron mechanisms may have important roles has begun to mount. Recent demonstration that flavins can facilitate electron acceptor usage (33) and that dopamine can be a respiratory electron acceptor (39) and past suggestions that cholesterol can also function in this capacity (40) indicate that we have much to learn about microbial adaptation to host environments. The human gut is not unique in its surplus of carbon and deficit of inorganic electron acceptors; environments such as salt marshes, peat bogs, and soils also have these characteristics (Fig. 1), highlighting that environments as disparate as the human gut and the bottom of a river provide similar physical, biological, and chemical drivers that select for novel solutions to the paucity of conventional electron acceptors.

One physical driver that can lead to small-scale changes in the oxygen environment is the movement of fluids. From the human tongue to a kelp forest, many systems have the potential to exhibit strong gradients in oxygen from boundary layer dynamics. Aquatic phototrophs rely on flow for the delivery of nutrients, and attributes of flow and the host phototroph can result in steep gradients and a diversity of microenvironments for oxygen. For example, oxygen and pH can change by a factor of 2 to 3 over a scale of 10 to 30 $\mu$m on the surface of a kelp blade (41); similarly, oxygen is quickly depleted in the top few millimeters of the air-sputum interface in samples taken from people with cystic fibrosis (CF) (42). The analysis of structures that break down a diffusive boundary layer, such as fine hairs, can lead to areas with differing concentrations of oxygen, suggested to be areas where epiphytic bacteria may colonize. In the brown alga *Fucus*, hyaline hairs increase the diffusive boundary layer, leading to nearly anoxic areas that might provide microsites for nitrogen fixation (43).

The feeding activities and other metabolic activities of animals can also lead to strong oxygen gradients that can change on the micron or millimeter scale, favoring different microbial metabolisms and taxa (44). In sponges, for example, feeding activities result in a 1-mm surface that is oxygenated, while the rest of the animal is anoxic, resulting in a set of predictable nitrogen dynamics associated with each (45). Inside the animal, nitrate reduction and metabolisms in the absence of oxygen occur, including anammox and denitrification, while the 1-mm oxygenated surface is a site for nitrification. Strong oxygen gradients in water column particles in the oxygenated ocean (46) are thought to provide novel low-oxygen microenvironments (47). The use of fiber optic oxygen microsensors will continue to be important to elucidate these micron- and millimeter-scale oxygen patterns.

Fluid movement in areas such as the human tongue may structure the microbial community through oxygen gradients in the same way that is suggested for aquatic plants and animals. Advances in imaging show both aerobic and anaerobic microbial taxa in close proximity and suggest that oxygen gradients determine the microbiome of the human tongue (48). The human gut also shows micron-scale changes in oxygen (49), suggesting that microbes that are hosted by humans reside across a diversity of oxygen environments, even though anaerobic processes dominate in the human gut. Biofilms, too, have pronounced changes in oxygen over micron scales (50). The human mouth can be an area of anaerobic respiration (51), and metagenomes suggest that

the anaerobic areas of the mouth are also associated with nitrate reduction, including denitrification (52).

Our understanding of human disease can be improved by attention to the redox state. The chronic infections that develop over decades in the airways of people with CF are associated with distinct redox changes. Impaired mucociliary clearance in CF leads to increases in airway fluids where microbial communities comprised of Gram-negative opportunistic pathogens such as *Pseudomonas aeruginosa*, along with oral anaerobes, develop (53). Measurements of CF sputum samples show that gradients in pH are steep and there are strong oxyclines, indicating a range of redox states (42, 54). Evidence for active microbial metabolism across the oxygen gradient has been accumulating, from fermentation products produced by facultative anaerobes found in CF breath samples (55) to the potential for nitrate reduction and an accumulation of ammonium (56) and nitrous oxide production indicative of denitrification (57). In the 1880s, the Ukrainian scientist Sergei Winogradsky put pond mud and water and several other resources for bacterial growth in a clear column (58). He observed strong stratification based on gradients of oxygen and other electron acceptors, and the method remains a way to demonstrate the stratification of taxa based on their affinity for specific electron donors and acceptors within redox gradients. While a Winogradsky column has been generally restricted to the study of aquatic ecosystems for over 100 years, cystic fibrosis sputum was recently set up in a similar manner. Again, strong oxygen gradients and their associated microbial metabolisms and taxa resulted (59), illustrating how the human host contains a range of microenvironments that may mediate infection.

Changes to the redox environment in the human gut often mediate dysbiosis and colonization by pathogens. When anaerobic microbial ecosystems are exposed to oxygen, bacteria related to opportunistic pathogens increase (60). The generally anaerobic human gut responds to antibiotic treatment with an increasing redox state that can deter a typical gut microbiome (61, 62) and encourage alternate microbial taxa that can tolerate antibiotic and redox stress and use different electron acceptors, including nitrate (35). Indeed, it is possible that an increased ratio of electron donors to electron acceptors may be an important determinant of a healthy human gut, though there is still much to understand about the ability of electron donors and acceptors and the role of external electron transport in the human gut.

With the advent of DNA sequencing methodologies, microbial taxonomy, component genes, and the probable function of those genes are increasingly described. With gene sequences in hand, there are efforts to predict biogeochemical function in the environment (25, 63), even in the absence of cultivated representatives of these taxa. Across taxa, core metabolisms primarily use six major elements and a set of known redox reactions. The interplay between elemental cycles, gene function, and taxonomic composition will be enhanced in human microbiome studies as cultivation and metabolic assays complement DNA sequence studies.

## CAN STOICHIOMETRIC APPROACHES PROVIDE INSIGHT ACROSS SYSTEMS?

The concept of stoichiometric limitation, where an element is in short supply relative to the availability of other elements, has a common application across systems and has provided a conceptual framework for the study of environmental microbes. In the ocean, the elemental ratio of a limiting nutrient, denoted $X$, to carbon (C) has shown that many elements can limit growth (64). The stoichiometry of nitrogen can control bacterial growth form, with the biofilm-forming mycobacteria existing as a solitary planktonic form when C/N ratios are low and then aggregating when carbon is increased relative to nitrogen (65). The ratio of $X$ to C can provide strong inference for the metabolisms that might be selected in a given environment, as well as the nutrient needs and constraints of hosts. For example, fish species show a tissue stoichiometry (C/N/P) that has much lower variance than gut contents, suggesting "homeostatic regulation" of elemental ratios (66). Nitrogen limitation in the mammalian gut has been proposed and tested, with a focus on the large intestine where nitrogen limitation might be particularly acute (67). Stable isotope analysis and imaging indicate that nitrogen limitation

can occur and is most acute for herbivores. For any consumer, from humans to aquatic invertebrates, the increased nitrogen that is needed may require an excess of carbon to obtain (66), a concept well demonstrated in aquatic plankton such as the water flea, *Daphnia* (68).

Because the components of DNA require nitrogen and phosphorus across all life forms, Elser et al. (69) suggest that the use of different nucleic acids and proteins might reflect elemental limitations, a concept they term stoichiogenomics. The stoichiogenomic perspective has some support from an analysis of the ubiquitous ocean cyanobacteria *Prochlorococcus*, where fewer atoms of nitrogen occur in the proteome in nitrogen-depleted waters (70). The application of a stoichiometric perspective has the potential to inform selection on microbial taxa in humans. Microbial proteins have different stoichiometry across human tissues, suggesting that the human host may limit the microbial community in different ways (71). Pathogenesis is often the result of a microbe harnessing a nutrient that would otherwise be in short supply (e.g., iron [72]) and relates to the study of "nutritional immunity" where hosts alter the availability and stoichiometry of transition metals (73). Indeed, stoichiometry and redox changes can disrupt redox homeostasis and lead to reactive oxygen or other reactive chemistry that can induce pathogenesis (74).

As carbon dioxide in the atmosphere increases globally due to the burning of fossil fuels, human plant food sources show decreased $X/C$ across a number of nutrients (75, 76), suggesting that humans are changing the elemental stoichiometry that we provide to our symbiotic gut bacteria. Elemental changes at the base of the food chain are further illustrated by century-scale increases in C/N across grassland plants sampled in herbaria (77). Across hosts, the stoichiometry of resource inputs compared with cellular machinery can reveal nutrient limitation and homeostatic regulation. While free-living bacteria in lakes show flexible and variable C/N/P (78), there is less understanding of bacterial stoichiometric constraints in association with hosts. Human tissues have distinct elemental stoichiometries that may be associated with growth rates (79), and some quickly growing cancers have high phosphorus due to nucleic acid production (80), raising important questions about whether stoichiometry could inform chronic infections. A key line of future investigation is needed to understand whether host-associated microbes exhibit elemental limitation or whether they alleviate stoichiometric limitations for the host. Including a quantification of nutrient dynamics in hosts may lead to a synthesis of the microbial taxa and metabolisms that might be selected across diverse environments.

## FUTURE RESEARCH CONNECTIONS ACROSS DIVERSE MICROBIOMES

As we increase our understanding of a diverse set of microbes in human health, we may become more predictive about the microbiome that can develop under different conditions, including how bacterial pathogenesis can be affected through alterations to the bio(geo)chemical environment (74). Proctor's (51) call for increased quantification of the environment in the human gut may be increasingly likely with the use of ingestible sensors for oxygen, hydrogen, and carbon dioxide (81). Whereas the "environment first" approach can reveal environmental states that favor particular metabolisms, the "microbe first" approach can detect bacterial genes that reveal important aspects of the environment. For example, Morris and Schmidt (82) searched for high-affinity terminal oxidase genes to indicate where microaerobic bacterial metabolisms exist. The possibility that microbial taxa serve as biosensors has rapidly become a reality.

The categorization of microbes into aerobes versus anaerobes has been a fundamental dichotomy for all disciplines that intersect with microbiology, and microbial study in humans has often focused on culturing and physically describing the microbial taxa. Although studies of human gut health indicate the importance of classifying aerobic and anaerobic bacteria and their interactions (83), an understanding of the redox environment across different human tissue types will likely enhance our understanding of pathogenesis and dysbiosis.

Finally, we recognize that anthropogenic activities have greatly increased the amount of three of the six major elements that microbes commonly use (carbon, nitrogen, and

phosphorus) while altering the availability of oxygen. The stoichiometry of energy sources is also changing. Climate change, in the form of increased heat in ecosystems, can increase periods of low oxygen, particularly in aquatic habitats. Experimental studies with the biofilm-forming *Roseobacter* showed an increase in biofilm formation as temperature increased and oxygen declined (84). Will continued heat stress in ecosystems lead to a shift in microbial metabolisms as a result of changes to the terminal electron acceptors that can be used in reactions? The dependence of electron donors and acceptors on redox states further portends alterations to microbial metabolisms across ecosystems.

## REDUCING BARRIERS TO SHARED APPROACHES

Researchers from different disciplines often have distinct vocabulary, practices, and traditions, and this is true for researchers in microbiome science across different systems. For example, researchers that focus on human microbiome research rarely are trained in biogeochemistry, a discipline that anchors the intellectual traditions of researchers working with the "environment first" approach to microbes. Biogeochemistry is often used to limit the range of metabolisms that can be energetically possible in environmental microbiology, as well as provide clues to metabolisms that have yet to be detected. The examples provided above from studies in lakes illustrate this line of inquiry. Outside the human body, there is a sustained tradition of measuring oxygen, pH, and nutrients, and Proctor (51) suggests that these determinants of microbial metabolisms should be measured more systematically in humans, though we acknowledge the practical and ethical challenges of measuring physical variables in human hosts. We emphasize that a knowledge of the redox environment is often inexpensive to obtain; the redox environment underpins all microbial metabolisms and should be a key component of research across systems.

In contrast, researchers working with host-associated and free-living microbes across diverse ecosystems have not relied on culturing to the extent that researchers have in human-hosted microbes. Recent estimates put the percentage of human gut microbes that have been cultured at 35% to 65% of those detected with DNA sequencing methods (85), though diversity may be underestimated (86). In contrast, the "microbe first"' approach relies on DNA and RNA sequencing techniques to infer taxa and function. Although this approach indicates that as much as 65 to 85% of environmental microbes remain uncultured (87, 88), the development of single-cell culture techniques for diverse microbes (87) will increase the discovery of microbial taxa and their metabolisms in systems outside the human host. We anticipate there will be an increased detection of the processes that structure metabolic capabilities across systems, leading to convergence in the understanding of community structure.

Symbioses and positive host-microbe interactions have characterized the study of nonhuman systems for decades, informing our understanding of foundational species such as coral reefs (89, 90), including the role of nitrogen fixation in essential crops recognized as long ago as 1880 (91). In contrast, the idea of symbiosis in humans is still nascent and is only recently generating research that informs possible symbioses in the human gut.

A principal barrier to sharing approaches in the discovery of environmental microbes and human-hosted microbes is the different training pipelines that are used for researchers in both areas. Solutions to connecting these distinct pipelines include convening cross-disciplinary workshops and ensuring interdisciplinary student training. The formation of microbiome centers around the country (92) and the world unite researchers across ecosystems and have membership across human and environmental systems, promoting idea and data sharing to enhance cross-communication (93). Through collaboration, we can determine the redox state for diverse microbial metabolisms, the electron acceptors, and electron donors that are associated with healthy versus pathogenic microbiomes and the common selective pressures that structure diverse microbiomes.

## ACKNOWLEDGMENTS

We thank J. Martiny and the Microbiome Center Consortium for facilitating this collaboration.

Conference funding for the MCC was provided to C. Pfister by the National Science Foundation (2002104), the National Institutes of Health (NCE R13 CA250289-01), the Office of Naval Research (N00014-20-1-2140), the Gordon and Betty Moore Foundation, and the W. M. Keck Foundation.

Comments from M. McFall-Ngai improved the manuscript. We thank B. Bays for Fig. 1.

C.A.P. was the principal author of the manuscript, with edits, commentary, discussions, and contributions from all authors.

We declare no competing interests.

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
