## [Reviewer comments · mSystems]

Conceptual exchanges for understanding free-living and host-associated microbiomes

Catherine Pfister, Samuel Light, Brendan Bohannon, Thomas Schmidt, Adam Martiny, Nicole Hynson, Suzanne Devkota, Lawrence David, and Katrine Whiteson

Corresponding Author(s): Catherine Pfister, University of Chicago

Review Timeline:

Submission Date:	November 13, 2021
Editorial Decision:	November 16, 2021
Revision Received:	December 15, 2021
Accepted:	December 15, 2021

Editor: Jack Gilbert

Reviewer(s): The reviewers have opted to remain anonymous.

Transaction Report:

DOI: <https://doi.org/10.1128/msystems.01374-21>

November 16, 2021

Dr. Catherine Ann Pfister
University of Chicago
1101 E 57th St
Chicago, IL 60637-1503

Re: mSystems01374-21 (Conceptual exchanges for understanding free-living and host-associated microbiomes)

Dear Cathy:

Thank you for submitting your manuscript to mSystems. We have completed our review and I am pleased to inform you that, in principle, we expect to accept it for publication in mSystems. However, acceptance will not be final until you have adequately addressed the reviewer comments.

Thanks for the article. In addition to myself one other editor read through the work, we only have a few small requests.

1. Why focus so much on human-associated microbiomes as contrasts to free-living microbes/microbial communities? Could you not replace human with host throughout the MS?
2. The figures could be aesthetically improved so that schematics replace the currently heavy text, and the visual focus is on the major differences between systems and opportunities for synergy.
3. This theme of "conceptual exchanges" in the title could be scaled to the 'exchange' of microbes to and from hosts/environments where the two are not inseparable from each other except for the relatively few strict host-associated microbes. For instance, how might cycling change for a microbe transferred from host to environment or vice versa? What are the adaptations that already exist for organisms to toggle their cycling in host and non-host associated environments.

Thank you for the privilege of reviewing your work. Below you will find instructions from the mSystems editorial office.

Preparing Revision Guidelines

Sincerely,

Jack Gilbert

Editor, mSystems

Journals Department
13 December 2021

Dear Dr. Jack Gilbert and mSystems editors:

Thank you for your rapid and thoughtful review of our ms "*Conceptual exchanges for understanding free-living and host-associated microbiomes*". We have considered the comments we received and we submit a revised manuscript and response to those comments below.

1. Why focus so much on human-associated microbiomes as contrasts to free-living microbes/microbial communities? Could you not replace human with host throughout the MS?

Our focus was on host-associated microbiomes and we mostly used that terminology throughout (L116-117), though our examples are very human-oriented. To better emphasize the generality of 'host-associated' to our readers, we have changed the text to 'host' in several places (L41 & L61). We note that the term 'host-associated' is now in the title, abstract, and multiple times in the Introduction, hopefully setting the stage. However, we do not want to lose our emphasis on human-microbe interactions because of the importance of reaching investigators in human-microbiome research, where culturing and other methodologies are proceeding rapidly; thus, we also point out different emphasis on host vs human (L89-90, L96-97).

2. The figures could be aesthetically improved so that schematics replace the currently heavy text, and the visual focus is on the major differences between systems and opportunities for synergy.

To improve the content of Figure 1, we took the time to reach out to a professional for design. The revised Figure 1 is a much nicer asset to the text. We also reduced some text in the manuscript (starting L164, starting L198) and tightened up the paragraph beginning L289.

3. This theme of "conceptual exchanges" in the title could be scaled to the 'exchange' of microbes to and from hosts/environments where the two are not inseparable from each other except for the relatively few strict host-associated microbes. For instance, how might cycling change for a microbe transferred from host to environment or vice versa? What are the adaptations that already exist for organisms to toggle their cycling in host and non-host associated environments.

This is an interesting point and one that we did not address previously. It also is addressed relatively little in the literature. We point this out where it comes up (L96-97) and revised the text starting L101.

The references are re-done because one reference was omitted in the edits above.

Thank you for the opportunity to make these revisions and we hope that we have thoroughly addressed the helpful comments that were raised. We look forward to hearing from you.

Sincerely,
Cathy Pfister
& on behalf of co-authors

December 15, 2021

Dr. Catherine Ann Pfister
University of Chicago
1101 E 57th St
Chicago, IL 60637-1503

Re: mSystems01374-21R1 (Conceptual exchanges for understanding free-living and host-associated microbiomes)

Dear Dr. Pfister:

Your manuscript has been accepted, and I am forwarding it to the ASM Journals Department for publication. For your reference, ASM Journals' address is given below. Before it can be scheduled for publication, your manuscript will be checked by the mSystems senior production editor, Ellie Ghatineh, to make sure that all elements meet the technical requirements for publication. She will contact you if anything needs to be revised before copyediting and production can begin. Otherwise, you will be notified when your proofs are ready to be viewed.

Publication Fees:

We recognize that the video files can become quite large, and so to avoid quality loss ASM suggests sending the video file via <https://www.wetransfer.com/>. When you have a final version of the video and the still ready to share, please send it to Ellie Ghatineh at eghatineh@asmusa.org.

Sincerely,

Jack Gilbert
Editor, mSystems

Journals Department
Phone: 1-202-942-9338